



# A study on the performance of low-cost sensors for source apportionment at an urban background site

Dimitrios Bousiotis[1], David C.S. Beddows[1], Ajit Singh[1], Molly Haugen[2],
Sebastián Diez[3], Pete M. Edwards[3], Adam Boies[2], Roy M. Harrison[1], and
Francis D. Pope[1*]

[1]Division of Environmental Health and Risk Management, School of Geography, Earth and
Environmental Sciences University of Birmingham, Edgbaston, Birmingham B15 2TT,
United Kingdom

[2]Department of Engineering, University of Cambridge, Trumpington Street, Cambridge,
CB2 1PZ, United Kingdom

[3]Wolfson Atmospheric Chemistry Laboratories, Department of Chemistry, University of
York, Heslington, York YO10 5DD, United Kingdom

*Corresponding Author, correspondence to Francis Pope f.pope@bham.ac.uk





## Abstract

While the measurements of atmospheric pollutants are useful in understanding the level of
the air quality at a given area, receptor models are equally important in assessing the
sources of these pollutants and the extent of their effect, helping in policy making to deal
with air pollution problems. Such analyses were limited and were attempted until recently
only with the use of expensive regulatory-grade instruments. In the present study we
applied a two-step Positive Matrix Factorisation (PMF) receptor analysis at a background
site using data acquired by low-cost sensors (LCS). Using PMF, the identification of the
sources that affect the air quality at the background site in Birmingham provided results
that were consistent with a previous study at the site, even though in different measuring
periods, but also clearly separated the anticipated sources of particulate matter (PM) and
pollution. Additionally, the method supplied a metric for the contribution of different
sources to the overall air quality at the site, thus providing pollution source apportionment.
The use of data from regulatory-grade (RG) instruments further confirmed the reliability of
the results, as well as further clarifying the particulate matter composition and origin.
Comparing the results from a previous analysis, in which a k-means clustering algorithm was
used, a good consistency between the results was found, and the potential and limitations
of each method when used with low-cost sensor data are highlighted. The analysis
presented in this study paves the way for more extensive use of LCS for atmospheric
applications and receptor modelling. Here, we present the infrastructure for understanding
the factors that affect the air quality at a significantly lower cost that previously possible,
thus opening up multiple new opportunities for regulatory and indicative monitoring for
both scientific and industrial applications.



## 1. Introduction

Air pollution is a major problem not only affecting human health (Pascal et al., 2013; Rivas et al., 2021; Shiraiwa et al., 2017; Wu et al., 2016; Zeger et al., 2008), but also causing environmental deterioration and social disparity due to its effect on climate change (Manisalidis et al., 2020; Mannucci and Franchini, 2017; Moore, 2009). This effect is more prominent especially within the urban environment or near pollution hot spots, though areas even hundreds of kilometres away from the emission sources can also be affected (Valavanidis et al., 2008, Bousiotis et al., 2021). As a result, the knowledge of the sources of air pollution is vital in both understanding the air quality at a given site as well as for policy making and action to improve air quality.  Such knowledge was provided, until now, by the analysis of data from expensive regulatory grade (RG) instruments, the use of which was not extensive due to their high cost and bulky size almost exclusively for scientific research. As a result, there is limited knowledge of the sources that affect the air quality. This is in part due to the exiguous deployment and spatial resolution of these expensive instruments (Kanaroglou et al., 2005), especially in low- and middle-income countries. In these areas the problem of air quality and its effect on human health is of great importance and expected to further increase in the coming years as a result of their rapid industrial and population growth (Kan et al., 2009; Petkova et al., 2013). To combat this, in the past decade, the development of low cost sensors (LCS) measuring either PM or gas phase pollutant concentrations has intensified (Lewis et al., 2018; Penza, 2019; Popoola et al., 2018), though still being far from an equal alternative to the more expensive RG instruments. Many limitations are associated with their use, with the main shortcoming being the inconsistency of their measurements, even for similar sensors deployed at the same site (Austin et al., 2015; Sousan et al., 2016), either due to operational and detector sacrifices that allow them to be inexpensive or from the effect of meteorological conditions that bias their measurements (Crilley et al., 2020; Hagan and Kroll, 2020; Wang et al., 2021). Thus, consistent calibration (Kosmopoulos et al., 2020; De Vito et al., 2020) and data corrections (Crilley et al., 2018; Liang et al., 2021; Vajs et al., 2021) are required for these sensors to provide reliable measurements (though sometimes even this is not enough) in addition to their continuous improvement and evolution (Giordano et al., 2021). Nevertheless, these

sensors have the potential to change the state of air pollution monitoring by allowing wider
use and better spatio-temporal coverage.
Many applications of LCS have been found in over the recent years providing measurements
at sites that were previously inaccessible by regulatory instrumentation, either due to being
economically difficult (Miskell et al., 2018; Omokungbe et al., 2020; Pope et al., 2018) or due
to the limitations set by their size (Jovašević-Stojanović et al., 2015; Nagendra et al., 2019,
Whitty et al., 2022). Additionally, the use of LCS made possible higher spatial resolution than
RG instruments (Feinberg et al., 2019; Krause et al., 2019; Prakash et al., 2021), greatly
improving the ability to measure air quality at more points of interest even at
neighbourhood scale (Schneider et al., 2017; Shafran-Nathan et al., 2019; Shindler, 2021),
supplementing the existing regulatory network (Weissert et al., 2020). While the
applications of LCS provided the information of the level of air quality at more sites, the vital
information of air pollution sources and the environmental conditions that enable or disable
air pollution, as well as their relative contributions is yet to be uncovered by their use. Pope
et al., (2018) using PM ratios managed to separate and identify the effect of major sources
of pollution in several cities in East Africa using data from LCS. Popoola et al, (2018)
identified the sources of pollution near Heathrow Airport, London using a network of LCS.
Bousiotis et al., (2021) using k-means clustering on PM data from both a LCS and an RG
instrument, showed the strengths and limitations of the sensor, in measuring particle
number concentrations and using them to identify the sources of pollution at a background
site in Birmingham, UK. While these studies identified many sources and conditions that
affected the air quality at the sites, there was no information on their temporal and relative
contribution.
In the present study, the two-step PMF (Beddows and Harrison, 2019), an advanced version
of a statistical method for source apportionment successfully applied in many studies with
RG instruments (Beddows et al., 2015; Harrison et al., 2011; Hopke, 2016; Leoni et al., 2018;
Pokorná et al., 2016), is applied on data collected from various LCS. This provides a
quantitative separation of the different sources and their contributions to a background site
located in Birmingham. Furthermore, data from RG instruments and an Aerosol Chemical
Speciation Monitor (ACSM) were also used in the analysis. This was done not only to
compare the results from the two sets, but to further characterise the sources of larger
particles at the site as well. The results of the present analysis are also compared with those
from a previous study at the same site made by Bousiotis et al., (2021), displaying the
additional information provided by the PMF as well as to check the consistency of the
results between the two methods. To the authors' knowledge source apportionment with
LCS data has only been attempted previously by Hagan et al., (2019) using Non-negative
Matrix Factorisation on a dataset from New Delhi, India, providing information of
combustion and non-combustion sources as well as their partial contributions in a three-
factor solution.  The present work sets the ground for future use of such sensors in a variety
of scientific and industrial scenarios, which can make feasible their wider use either as
standalone sources of the data needed for such studies or in combination with RG
instruments for better spatial coverage.

## 2. Methods

### 2.1 Location of the site and instruments

The measurement site is the Birmingham Air Quality Supersite (BAQS) located at the
grounds of the University of Birmingham (52.45°N; 1.93°W) (fig. 1). This is an urban
background site within a large residential area about 3 km southwest of the city centre of
Birmingham. For this site, PM concentration measurements in the range 0.35 to 40 µm were
collected using an Alphasense OPC-N3 in a 10 second resolution (averaged in 1-hour
resolution) for the period between 16/10/2020 to 30/10/2020. Additionally, data from
several LCS were also collected. $NO_x$ and ozone measurements were collected using the Box
Of Clustered Sensors (BOCS, Smith et al., 2019) in the same time resolution, as well as black
carbon (BC) concentrations using the MA200 sensor by Magee Scientific. Finally, the data for
the lung deposited surface area (LDSA) of particles in the range of 10 nm to 10 µm, which is
found to strongly correlate with BC emissions (Lepistö et al., 2022), was collected using a set
of two Naneos Partectors by Naneos Particle Solutions GmbH. One sensor measured the
surface of all particles in this size range, while the second is placed after a catalytic stripper
(Catalytic Instruments CS015) which removes the semi-volatile particles (Haugen et al.

131    2022).

Apart from the data provided directly from the sensor before the catalytic stripper, the ratio
between the measurements of the two Naneos Partectors was also considered according to:





$$LDSA_{ratio} = \frac{LDSA\ after\ the\ catalytic\ stripper}{LDSA\ before\ the\ catalytic\ stripper}$$

This was done to resolve whether such a configuration can provide additional information
for the origin of pollution or the age of the pollutants in the incoming air masses, as
increased concentrations of semi-volatile compounds are usually associated with
anthropogenic sources, especially in the urban environment (Mahbub et al., 2011, Schnelle-
Kreis et al., 2007, Xu and Zhang, 2011). Thus, a high $LDSA_{ratio}$ is expected to be associated
with fresher pollution which usually has a higher content of volatile compounds (i.e.,
pollution sources at a close distance from the site), while lower ratios are probably
associated with either cleaner conditions or more regional and aged pollution with higher
concentrations of semi-volatile compounds, generally associated with sources at a greater
distance from the measuring site. This specific metric was also used in our previous study
(Bousiotis et al., 2021) and the consistency of the results between the two will be
compared.
For better characterisation of the larger particles, the Aerodyne ACSM was used, providing
information about its composition in the size range between 40 nm to 1 μm for $NO_3^-$, $SO_4^{2-}$
and organic content. For the comparison of the results, data from RG instruments were also
used, namely a Palas FIDAS (for PM), a Teledyne T500U (for $NO_x$), a Thermo 49i (for $O_3$) and
an AE33 aethalometer from Magee Scientific (for BC). Comparison of the regulatory
instruments and the LCS allows for consistency of the results between instrument types to
be checked. More information about the sensors and instruments used in this study can be
found in Bousiotis et al., (2021).
Finally, for the present study the PMF analysis was performed using the second iteration of
the PMF software developed by Paatero (2004a; 2004b). Data was analysed using the
Openair package for R (Carslaw and Ropkins, 2012), and back trajectory data were
extracted by NOAA Air Resources Laboratory and calculated using the HYSPLIT model
(Draxler and Hess, 1998).


**2.2 Positive Matrix Factorisation**


The PMF is a multivariate data analysis, developed by Paatero (Paatero and Tapper, 1993;
1994), which is the most commonly used method for source apportionment and has been
applied numerous times in the field of aerosol science. The method is a weighted least-
squares technique that describes relationships among species measurements (Reff et al.,
2007). It assumes that X is a matrix of observed data, typically either particle number size
distributions (PNSDs) or chemical composition data, and u is the known matrix of the
experimental uncertainty of X. Both X and u are of dimensions $n \times m$ (where n is the number
of measurements and m is the number of species measured). The method solves the
bilinear matrix problem X = GF + E where F is the unknown right hand factor matrix
(sources) of dimensions $p \times m$, G is the unknown left hand factor matrix (contributions) of
dimensions $n \times p$, and E is the matrix of residuals. The problem is solved in the weighted
least-squares sense: G and F are determined so that the Euclidean norm of E divided
(element-by-element) by u is minimized. Furthermore, the solution is constrained so that all
the elements of G and F are required to be non-negative (Paatero and Tapper, 1994). Higher
F values account for better association of the given variable with the factor it is assigned to,
while higher G values account for greater contribution of the factor at the given time period.
In the present analysis, a combination of both PNSD and particle composition data were
used. Such a combination may cause several shortcomings in the application of the PMF as
different types of data are used (Beddows and Harrison, 2019). To overcome these
shortcomings the two-step PMF method, proposed by Beddows and Harrison (2019), was
used. In the first step of the method, a part of the dataset is PMF-analysed (i.e. composition)
and a solution is provided. The time series G values (and errors) of the solution from the
first step are then used as input variables to the second step, where they are combined with
the additional measurements (i.e. PNSD data) dataset applying a second PMF analysis. In
the present study the opposite path was considered, with the first step using the PNSD
provided by the OPC sensor and the inclusion of particle composition data in the second
step. This was explicitly done for two reasons: 1. to test the capabilities of the LCS in source
apportionment, 2. to connect specific PNSD profiles with specific pollution sources.
Furthermore, on the second step of the analysis detailed in Beddows and Harrison (2019)
the explained variance of the factors from the first step were maximised. This directly
connects the additional variables in the second step with the PNSD profiles found in the first



step, excluding the possible factors formed with the data from the additional LCS data. In
the present study, this step in this method was omitted, as the aim is to present the results
of the receptor model as they occur in real life using a combination of LCSs measuring both
particle number concentrations and composition.
For the study site, particle number concentration data were available from the OPC for
particles of diameter < 40 μm, but only data up to 10 μm were used. This was due to the
lack of sufficient non-zero counts in the larger size bins above that size threshold, which
disfavours PMF analysis to be completed. Additionally, separate LCS data for NO and $NO_2$
were available. The NO data showed sensible variation, however, a great number of the NO
data points had low negative values due to their very low concentrations, which is
impossible data for the PMF algorithm. Rather than removing the negative numbers or
artificially calibrating the data upwards, we use NOx (NO + $NO_2$) as the variable of interest.
Finally, to avoid the increased uncertainties from the use of unavailable data (as missing
data are treated with increased uncertainties), a time window for which all data were
available was chosen. Thus, data availability is 100% and no special treatment was
considered for missing data.

## 3. Results

### 3.1 General conditions at the BAQS site.

The measuring period (16[th] to 30[th] of October 2020) was chosen as it is a period which
presented rather typical meteorological conditions in the area, had no missing data from
any of the instruments used, and because they were the last days before the second
lockdown due to COVID-19 was applied (31[st] of October 2020). General meteorological
conditions were rather typical for the period in Birmingham, UK. As a result, the conditions
and activities in the surrounding area found in this period are considered almost consistent
with the normal conditions at the site in the autumn season. Mean temperature was 10.0 ±
2.5°C and mean relative humidity was 87.9 ± 7.5 % (standard deviations are calculated using
hourly data) during the measurement period. The average wind profile (Fig. S1) was also
typical for the UK with mainly southwestern winds of relatively low speed (2.1 ± 1.1 m s⁻¹).

### 3.2 First step PMF analysis (PNSD analysis)

PMF is a descriptive model having no objective criterion in the choice of the optimal number of factors (Paatero et al., 2002). A 4-factor solution was chosen for this analysis. This is due to the relatively limited period analysed as, as mentioned earlier, no significant variation was found in either the meteorological conditions or the sources that affected the air quality in the area. Solutions with additional factors were also attempted but these provided no extra information on additional sources, rather the additional factors separated factors that had already found into smaller groups with no significant covariation. The PNSD profiles of the factors found are presented in Figure S2. Due to the limited variation of the PNSD profiles when presenting all the size bins available, making some of them appear identical (i.e. Factor 2 and 3, due to the increasing particle number concentration as the size decreases), the smallest particle diameter size bin at 400 nm (particle diameter range between 350 to 460 nm) was removed to better present the variation on the larger sizes. Thus, the particle profiles without the smallest available size are presented in Figure 2. The profiles in the range between 500 nm to 10 μm for the four factors, associated with unique formations extracted from the method are:

- Factor 1, that presents no significant peaks in the measured range of the OPC, but does show a steady increasing trend with particle diameters below 1 μm
- Factor 2, with a distinct particle diameter peak at about 2 μm
- Factor 3, with a distinct particle diameter peak at about 2 μm and an increasing trend below 750 nm
- Factor 4, accounting for particle diameter peaking at about 750 nm and 1.5 μm.

### 3.3 Second step PMF with LCS data (LC analysis)

The four-factor solution was also chosen in the second step analysis, for which the results of the first step are combined with the additional particle and gas phase composition datasets from LCS. The addition of more factors instead of adding information or providing clearer associations with the factors from the first step, it separated the existing factors and their association with the particle composition data into mixed factor groups with less significant contributions of the variables. The association of the variables with each factor is presented in figure 3, while the temporal variation of the contributions G of all the factors from this


analysis is presented in figure 4, along with the wind profile for some periods when each
factor was dominant.

The four new factors are:
**LC1 (Local and city centre pollution on calm conditions)**: The LC1 is strongly associated with
the first factor from the initial PMF on the PNSD. For the period when the contribution of
this factor is higher ($18^{th}$ and $19^{th}$ of October, see fig. 4) rather slow winds prevail from
many sectors (in this case mainly from the southwest). This factor has higher contributions
during calm conditions and during periods with north-eastern winds, though with lower
contribution (Fig. 5). It is highlighted that at the northeast of the specific site is the city
centre of Birmingham which is one of the main sources of pollution as found from a
previous study (Bousiotis et al., 2021). Looking at the diurnal variation (Fig. S3) of this factor
we see increased contributions during early morning and evening hours, likely associating it
with the morning and evening rush hours. The increased contributions during night-time
should not be overlooked and are probably the result of the lower boundary layer height
(BLH) during this time of the day. Additional data analysis shows an increased association of
this factor with $PM_1$ (Fig. 3), though this association is reduced for particles of larger sizes,
further confirming the lack of additional peaks on greater sizes. This along with the
increased association with the LDSA indicates the presence of large number of particles
below the detection limit of the instrument. This factor is also associated with almost all the
pollutants used, such as $NO_x$, CO and BC, though not as strongly as factor LC3 that is
discussed below, probably associated with pollution sources in a closer range to the
measuring station, as well as to a smaller extent with pollution from the city centre. Its
connection with air masses from the northeast is also confirmed from the back trajectory
analysis (Fig. 6), in which the highest contributions of this factor were found for air masses
from the northeast.
**LC2 (Marine)**: This factor is strongly associated with the fourth PNSD factor from the initial
analysis (fig. 3). It presents relatively high association with PM which increases as the size
increases. No other significant association is found rather than relatively weak ones with
ozone, CO and the LDSA$_{ratio}$. It does not have a clear diurnal variation (fig. S3), though it has
slightly increased contributions during night-time. Higher contributions for this factor are
found with south and south-eastern winds of high speed (fig. 4 and 5). This can be seen in



Figure 4, where the highest contributions of this factor are associated with strong southern
winds. The marine nature of this factor is clearly highlighted through the back trajectory
analysis for this factor (Fig. 6) in which higher contributions are mostly found with air
masses originating from the north Atlantic Ocean, while some contributions from southern
Spain and Africa, which may be associated with Saharan dust and pollution from these
areas.
**LC3 (midday city centre and southwest pollution)**: This factor does not have any significant
association with any of the factors from the PMF analysis of the PNSD (fig. 3). It presents
greater contributions during the midday (fig. S3), and it is associated with north-eastern and
southwestern winds (fig. 5). It has high contributions with all the pollutants included in the
analysis and the LDSA$_{ratio}$, which points to fresher pollution (pollution sources closer to the
measuring station). Such sources of pollution in most cases are associated with particles of
sizes smaller than that measured by the OPC, hence the lack of association with any of the
factors found from the PNSD analysis. The back trajectory analysis provides no clear origin
for the air masses of this factor (fig. 6), which may indicate a relatively smaller pollution
lifetime, which is associated with incoming air masses from all directions.
**LC4 (Urban background)**: This factor has a rather strong association with the second factor
from the PNSD analysis and a weaker one with the third one (Fig. 3). It does not have a clear
diurnal variation (fig. S3) and it is mainly associated with north-eastern winds (Fig. 5). It
presents weak associations with all the variables inputted in the PMF analysis making it hard
to distinguish either a source or conditions for which this factor is enhanced. The back
trajectory analysis though shows that this factor is associated with air masses from
continental Europe as well as Scandinavia (Fig. 6), which for the UK, usually contain aged
and hence typically larger secondary PM pollutants.

**3.4 Second step PMF with RG data (RG analysis)**
While the primary aim of the present study is to highlight the capabilities of LCS in source
apportionment, the measurements provided by these devices are mainly focused on gas
phase pollutants which are in most cases associated solely with ultrafine particles. The OPC
measurements used for this site have a particle diameter range between 400 nm to 10 µm.
Thus, apart from using data from RG instruments measuring gas phase pollutants, it was



considered sensible to add data from an ACSM, which measures compounds associated with
larger particles, such as nitrate, sulphate, and organic compounds (used in this analysis).
Some of the factors in this analysis are rather similar with those formed from the analysis
using LCS dataset. Thus, the **RG1** factor in this analysis is mainly associated with the first
factor from the PNSD analysis in the first step (Fig. 7), similar to that found also in LC1 (Fig.
3). The wind conditions are also similar for which these factors from the two analyses
present their highest contribution (Fig. 8), as well as their temporal variation (Fig. S4) and
diurnal variation (Fig. S5). The additional information granted using the ACSM data is the
strong association of this factor with nitrate, and a stronger association with $NO_x$ and BC are
also found, compared to the LC analysis. This further associates this factor with nearby
sources of pollution which prevail with low wind speeds and may associate the conditions of
this factor with the low BLH height found during that time, though high contributions were
also found for early morning and evening hours, as in the LC analysis for the similar factor.
Finally, the back trajectory analysis (fig. 9) shows higher contributions associated with air
masses from the northeast, further confirming its similarity with the first factor from the LC
analysis and its urban origins.
The **RG2** is unique and has no association with the factors from the PMF on PNSD data and
is strongly associated only with sulphate (Fig. 7). It does not have a clear diurnal variation
(fig. S5) and seems to have higher contributions with southwestern winds of rather high
speed and to a lesser extent with north-easterly winds (Fig. 8). The back trajectory analysis
(Fig. 9), while presenting few relatively high contributions from continental Europe, mainly
associates this factor with incoming air masses from all sea origins surrounding the UK. This
is expected as the ocean is a source of sulphate containing compounds (for the particles at
the size range measured by the OPC), either sea-salt sulphate or marine biogenic sulphate
(Lin et al., 2012; Raes et al., 2000).
The **RG3** is similar to the LC2 and is mainly associated with the fourth factor from the PNSD
analysis and to a lesser extend with the third (Fig. 7). This factor has slightly increased
contributions during night-time (Fig. S5) and south and southwestern winds (Fig. 8). It
presents increased associations with increasing PM size, though in this case it is also
strongly associated with $O_3$. Unfortunately, no Cl or Na data were available to further
determine the marine nature of this factor. The back trajectory analysis though once again
presents higher contributions with marine air masses (Fig. 9), though some hot spots are


also found from continental Europe, which probably explain to an extent the small
associations found with $NO_x$ and organic compounds from the ACSM.
Finally, the **RG4** is mainly associated with the second factor and to a lesser extent with the
third from the PNSD analysis (Fig. 7). It presents higher contributions with north-eastern
winds (Fig. 8), has an unclear diurnal variation (Fig. S5), and presents higher contributions
with air masses from continental Europe (Fig. 9), like the LC4 from the second-step analysis.
While in that analysis it was difficult to characterise the sources for that factor, the strong
association with organic compounds found here with the addition of the ACSM data helps in
its clearer characterisation.

## 4. Discussion
### 4.1 Comparison of the results from the second-step analysis
It should be noted that regardless of any possible similarities between the two (second-
step) analyses, a direct comparison of the results should be conducted with great care. As
different variables are considered, even minor differences may result in different trends,
contribution of variables and the sources described. Regardless, the results of the two
analyses have great similarities especially on specific factors that are associated with the
same particle size distribution profiles (from the PNSD analysis), contribution of chemical
compounds and diurnal variation. Three factors were found to have great similarities and
were associated with similar particle profiles. Specifically, these are the factors describing
the sources of particles which are either in close proximity to the measuring station or occur
with almost calm conditions (Factor 1 on both analyses), the marine factor (Factor 2 on LC
analysis and 3 on RG analysis) and the continental factor (Factor 4 on both analyses).
Looking at their temporal contributions (Fig. 4 and S4), the first factors on both analyses
appear to consistently peak on periods when the second set of factors (LC2 and RG3)
presents lower G contributions (and vice versa), which is expected due to the nature of their
sources. The factors on both sets though have almost identical temporal variation of their G
contributions regardless of the dataset. For the fourth factors on both analyses, though
presenting similar associations with their variables, differences are found in their temporal
variations with the addition of the ACSM data. This shows that while these factors appear to
be almost identical, small differences can still be found in their temporal variation and
variable associations, when different datasets are considered. Nevertheless, the addition of
the ACSM data shows a very high contribution of $NO_3^-$ on the first RG factor, $SO_4^{2-}$ for the
second factor and the organic component on the fourth factor.
The remaining factor from both analyses though is completely different between the two
analyses and point towards the differences on the variables used for each. In the LC analysis
the factor formed consists of sources that are associated with fresher pollution sources.
Thus, a factor with strong associations with all the pollutants available was formed, it was
not associated with any of the PNSD formations from the first-step analysis and presented a
unique diurnal variation peaking midday. This should be expected as the particle size
measured by the OPC is much larger compared to the size of the particles these chemical
compounds are usually associated with. The occurrence of this factor was probably included
partially to the first and fourth factor of the RG analysis, as these present relatively higher
associations with $NO_x$ and BC and more enhanced contributions during midday hours
compared to their LC analysis counterparts.
Finally, using the RG instrument data, the additional factor is associated with sulphate
alone. This is a result that was consistent regardless of the number of factors used, either
greater or smaller. Sulphate containing compounds have a lower volatility compared to the
other chemical compounds used in the analysis and is relatively more stable with a rather
small seasonal variation (Utsunomiya and Wakamatsu, 1996), thus having a longer lifespan
and distance of travel. As a result, sulphate was found not to be associated with any other
chemical compound and always formed a factor of its own (regardless of the number of
factors chosen).

**4.2 Comparison with the results from a previous study.**
Although different methodologies were used with the previous analysis for the BAQS site
(Bousiotis et al., 2021), as well as for different time periods, many similarities were found
for the sources of particles at the site. The main source of smaller particles at the site in the
previous analysis is found to be the city centre in the northeast, for which relatively high
concentrations of $NO_x$ were found. Similar is the case in the present analysis, as for the
sources found to be associated with north-easterly winds an association was also found with
$NO_x$ and the $LDSA_{ratio}$. Additionally, a source of sulphate found with southerly winds was also
confirmed in the present study, with the association of high sulphate concentrations with a
factor, which presents higher contributions with winds from the southern sector. While in
the previous analysis the sources responsible for this source could not be pinpointed, in the
present analysis, using a back trajectory analysis, the sulphate factor was associated with
marine particle sources from all directions. Furthermore, a factor in the present analysis,
which identifies hot spots south of the measuring station with strong presence of PM of all
sizes, was also found with the k-means analysis in the previous study, though in that case it
was more associated with the pollution sources from that side rather than the long-range
transport found here.
These similarities are very encouraging, as even though the analyses were made for
different periods and using different methods, there is consistency between the results. This
means that regardless of the different seasons studied (previous analysis was performed
during winter to early spring), the sources of particles (and pollution) are relatively uniform,
without significant changes.
Additionally, the k-means method identified sets of conditions that either promote or
supress the pollution at the sites (as this can be illustrated with the variable particle
concentrations between the clusters found from the analysis), rather than separate sources
of pollution that affect the site. While this provides a more realistic picture of the conditions
it makes it harder to distinguish the specific sources and their effect in its air quality. On the
other hand, the PMF not only provides clearer separation of the sources, but the temporal
contribution of each source as well, which shows the real extent of the effect of each source
of particles or pollutants, thus achieving source apportionment rather than just the
identification of pollution sources that the k-means offers. The k-means approach identifies
the effect of the sources of particles, but it also separates cleaner periods as separate
clusters. These two effects gives a more complete overall picture of the air quality at a site.
PMF could also provide this information, but it would be more difficult to obtain looking at
the different sources and the conditions that keep them to low contributions (this would
also require a much greater number of factors).
Furthermore, due to the complexity of the clusters from the k-means, pinpointing the
sources that the particles are associated with is difficult. This is due to the clusters, being a
set of different sources and conditions rather than clearly separated sources, were not
clearly associated with distinct wind directions, speeds or hot-spots. Contrary to that, the



factors formed by the PMF present clearer association with specific sectors, thus making it
easier to define the sources associated with them, as in the results they are presented as
hot spots within the polar plots.
The analysis of atmospheric data using either k-means or PMF are proven to provide
adequate and trustworthy information for the sources of particles and by extension of
pollution at a site, even with the sole use of LCS as shown in this paper and the preceding
Bousiotis et al. 2021 paper. The combined use of both approaches provides a clearer picture
of the different sources and their effect, as the PMF is able to better separate and provide
the effect of the sources of pollution that affect the air quality at a site and the k-means
provides a more realistic representation of the conditions at a site, by showing the
combined effect of these sources. The relative consistency of the results found between the
two analyses, even being in different time periods, is very encouraging and shows that the
very important information of pollution receptor modelling is viable with LCS, providing a
much-needed alternative for countries or scenarios where the use of regulatory-grade
instruments is not feasible. The significantly lower price point of LCSs means that in addition
to hyperlocal measurement of air pollution, it should now be possible to deliver hyperlocal
source apportionment of air pollution. This ability will open new research and industrial
abilities to pinpoint air pollution sources and subsequently manage them.
Finally, the LDSA$_{ratio}$, a variable that was introduced in the previous analysis, was included in
the present one as well. As in the previous analysis, this ratio was found to be more
associated with fresher pollution from combustion sources near to the measuring station,
for which it has reliably performed in both analyses.

## 5. Conclusions

To solve air quality problems and to deliver the associated policy making effectively, it is
vital to have a methodology to measure the sources of air pollution, and their relative
importance. Historically, this has been achieved using expensive RG instruments. The cost
implications of these studies make assessment at dense spatial resolutions limited. In this
study, data from a low-cost OPC and other LCS, measuring gas phase pollutants, black
carbon and the lung deposited surface area of particles in BAQS were analysed using the
two-step PMF analysis. Four factors were formed from this analysis and were associated





with their respective sources and to a great extent with unique PNSD profiles. The following
factors were found: a factor associated with either combustion sources in close proximity of
the measurement site or associated with calm conditions, a marine factor, a factor
associated with midday activities from the city centre and a more constant factor from the
northeast. The same analysis was also performed using data from RG instruments and the
same PNSD factors. This was done to evaluate the results from the low-cost sensor analysis,
as well as to further characterise and clarify the sources associated with the factors formed.
Significant agreement was found between the results of the two analyses, highlighting that
the LCS are capable for carrying out such analyses. The additional ACSM data from the
second analysis further helped in the characterisation of the composition of the particles of
each factor, clarifying the sources associated with nitrate, sulphate and organic compounds
at the site, as well as strongly associating some with unique PNSD profiles. While in their
present state, the LCSs do not possess the full capability of the RG instruments for providing
high accuracy measurements, considering the limitations they were found to be adequate in
providing with the trends of the particles and pollutants measured which are important for
source apportionment studies. This is done at a fraction of the equipment cost; see
Bousiotis et al. 2021 for cost estimates.
Furthermore, comparing the results from the PMF to those from the k-means analysis
showed the different strengths and weaknesses of each approach. The PMF is better in
pinpointing the effect of separate sources of pollution, but it is difficult to give a clear
representation of the actual conditions when each factor affects the site. The k-means is not
as efficient in clearly separating the different sources, but it does provide a more realistic
picture of the air quality at a site in relation to the ambient conditions. The combined use of
both methods though provided a clearer picture for the conditions at the site.
The methodologies developed and used in this study will help to reliably facilitate source
apportionment studies in the future, with either the sole use of LCS or their combination
with RG instruments. As for a given site, specific PNSD formations are associated with
specific conditions and sources (Harrison et al., 2011), by creating a repository of unique
PNSDs at a site and associating them with their respective sources, in the future the source
apportionment may be done to an extend using only PNSD profiles and meteorological data
alone. This will do much in simplifying the source apportionment process allowing its wider
application and help in dealing with environmental challenges. For this though, further



testing in more diverse environments and scenarios is needed which, along with the
anticipated development of the LCS, will provide a denser and reliable measuring network
even for countries with lower incomes and help for cleaner and healthier environmental
conditions.


**Author Contributions**
The study was conceived and planned by FDP who also contributed to the final manuscript,
and DB who carried out the analysis and prepared the first draft. AS, MH, DCSB and SD
provided data for the analysis. DCSB provided help with the analysis of the data. RMH, PME
and AB contributed to the final manuscript.

**Competing Interests**
The authors have no conflict of interests.

**Acknowledgements**
We thank the OSCA team (Integrated Research Observation System for Clean Air) at the
Birmingham Air Quality Supersite (BAQS), funded by NERC (NE/T001909/1), for help in data
collection for the regulatory-grade instruments. We thank Lee Chapman for access to his
meteorological dataset used in the analysis.
**Financial support.**
This research has been supported by the Natural Environment Research Council (NERC grant
no. NE/T001879/1), the Engineering and Physical Sciences Research Council (EPSRC grant
no. EP/T030100/1) and internal EPSRC funding provided to the University of Birmingham for
Impact Acceleration.




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





**FIGURE LEGENDS**

**Figure 1:**    Map of the measuring station.

**Figure 2:**    Particle profiles of the factors from the PMF analysis (> 500 nm). The lines
indicate the average particle count per second for each particle size bin.


**Figure 3:**    Variable association for the factors from the LC analysis. Grey bars indicate
the values of F, while red bars indicate the explained variations for each
variable.


**Figure 4:**    Temporal variation of the contributions of the factors from the LC analysis. The
windroses refer to the wind conditions for the corresponding periods when
specific factors presented higher G contributions.


**Figure 5:**    Polar plot of the average G contributions of the factors from the LC analysis.

**Figure 6:**    Average G contribution of the factors from the LC analysis for incoming air
masses. Higher contributions indicate better association of the given factor
with the corresponding air mass origin.


**Figure 7:**    Variable association for the factors from the RG analysis. Grey bars indicate the
values of F, while red bars indicate the explained variations for each variable.


**Figure 8:**    Polar plot of the average G contributions of the factors from the RG analysis.

**Figure 9:**    Average G contribution of the factors from the RG analysis for incoming air
masses. Higher contributions indicate better association of the given factor
with the corresponding air mass origin.






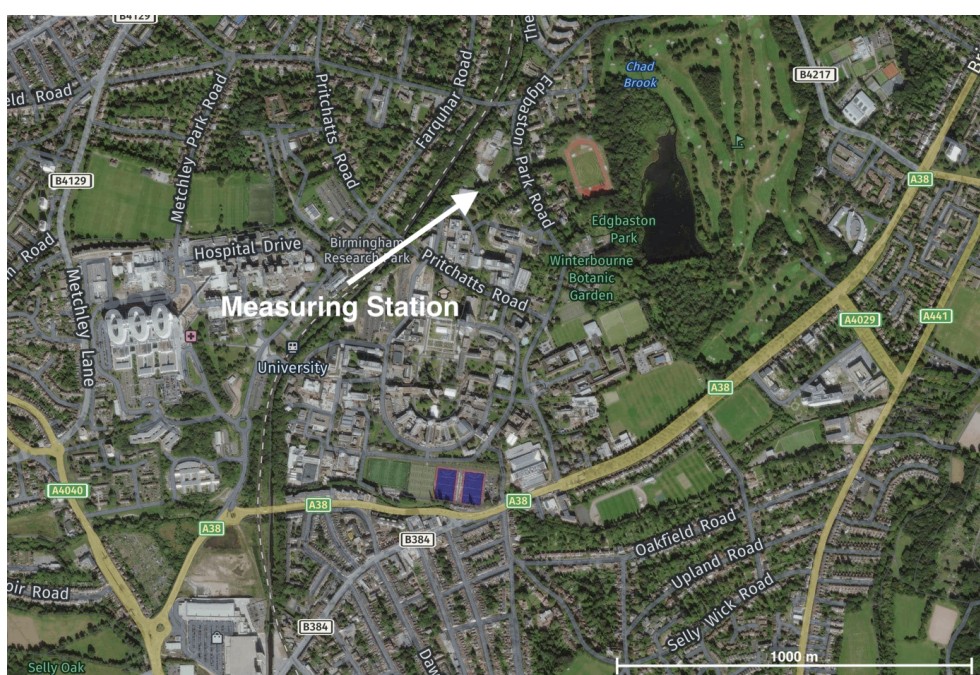


Figure 1: Map of the measuring station. Imagery @2022 Bluesky, Getmapping plc, Infoterra
Ltd & Bluesky, Maxar Technologies, The GoeInformation Group, Map data
©2022







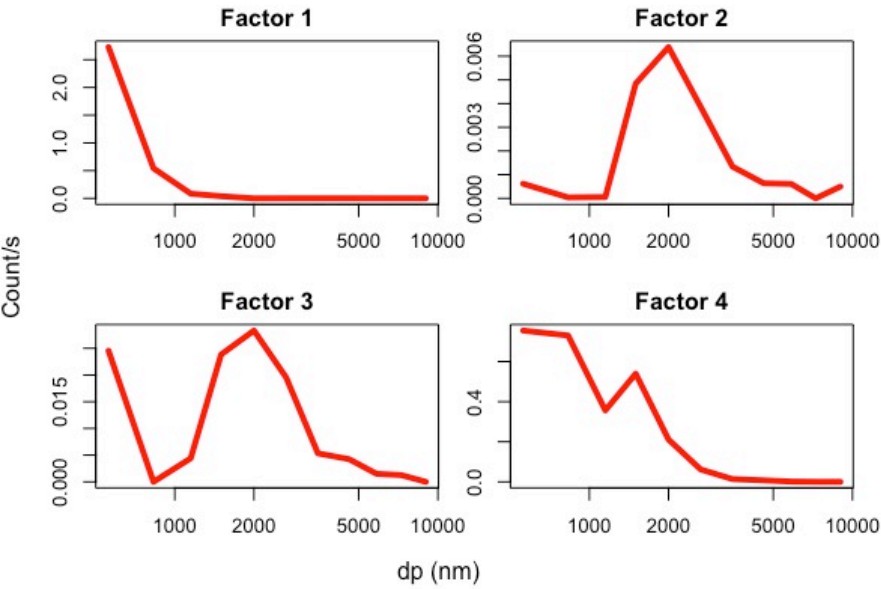


Figure 2: Particle profiles of the factors from the PMF analysis (above 500 nm). The lines
indicate the average particle count per second for each particle size bin.






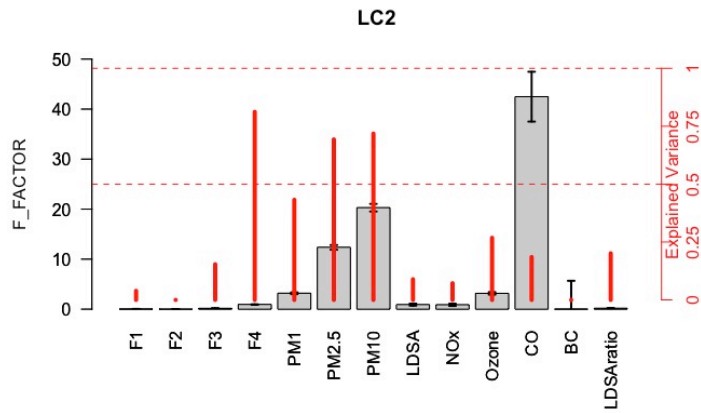


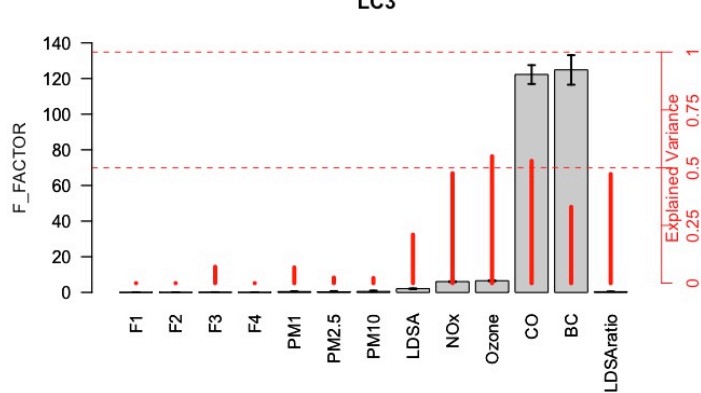


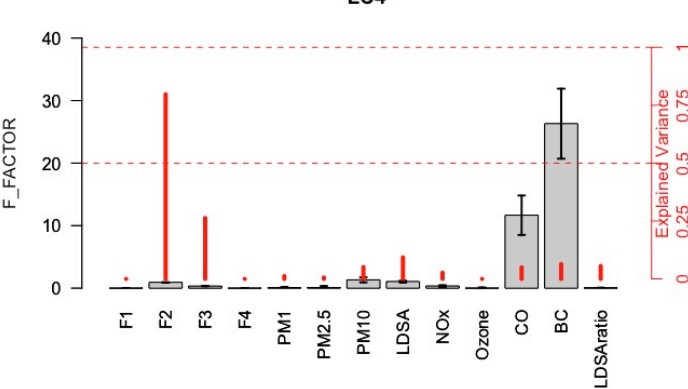


Figure 3: Contribution of the factors from the LC analysis. Grey bars indicate the values of F,
while red bars indicate the explained variations for each variable.

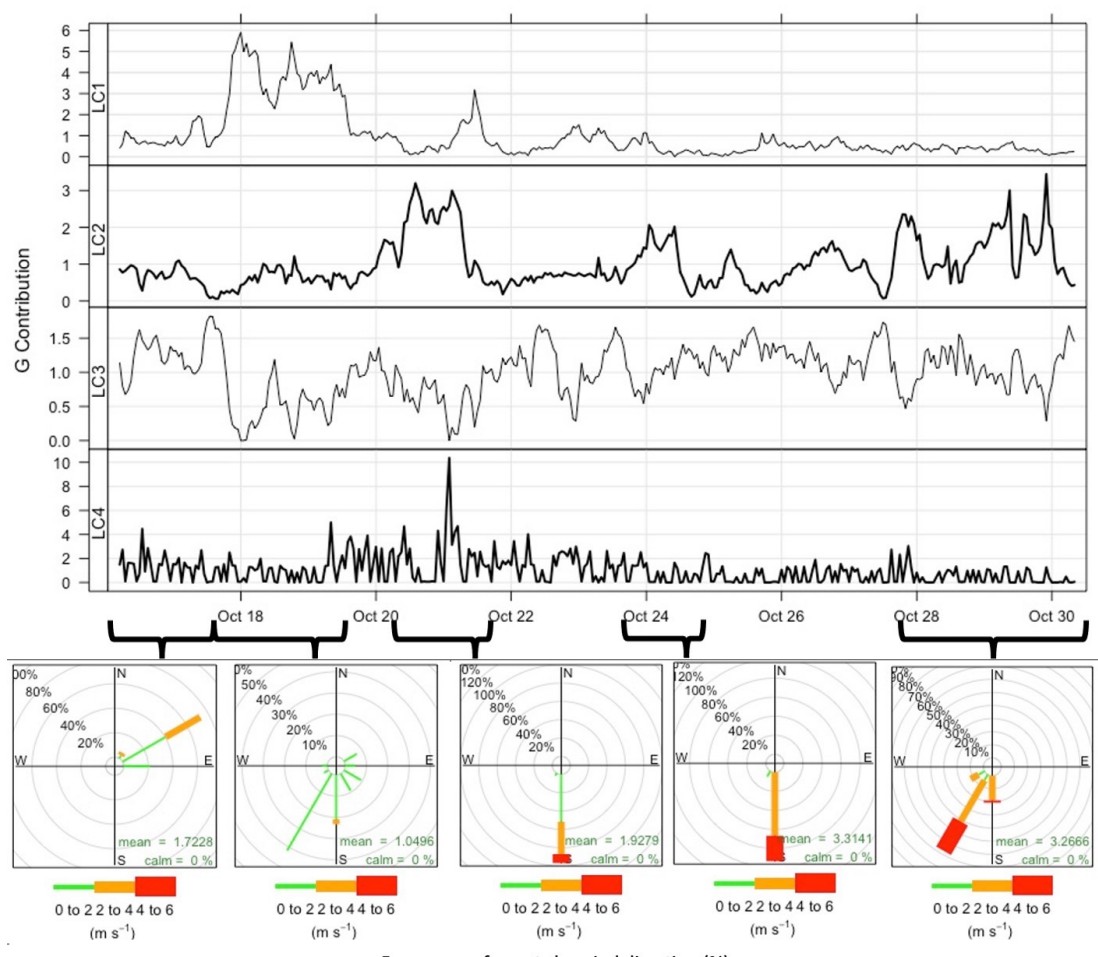

Figure 4: Temporal variation of the contributions of the factors from the LC analysis. The windroses refer to the wind conditions for the corresponding periods when specific factors presented higher G contributions.



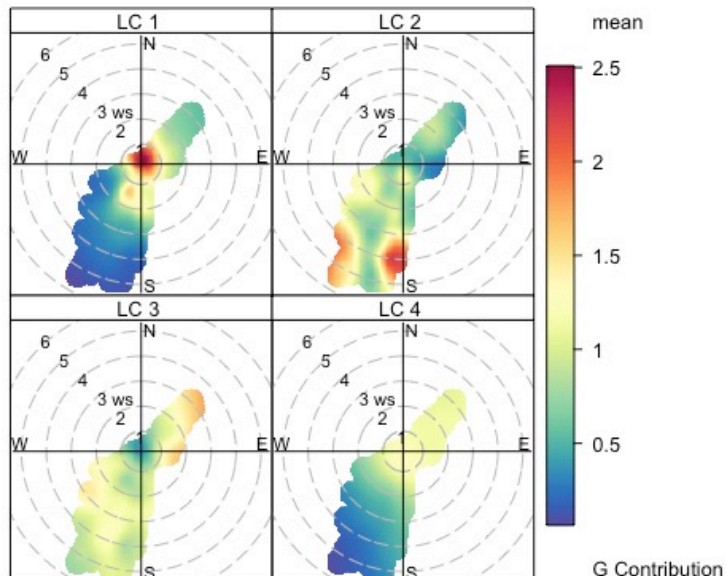


Figure 5: Polar plot of the average G contributions of the factors from the LC analysis.



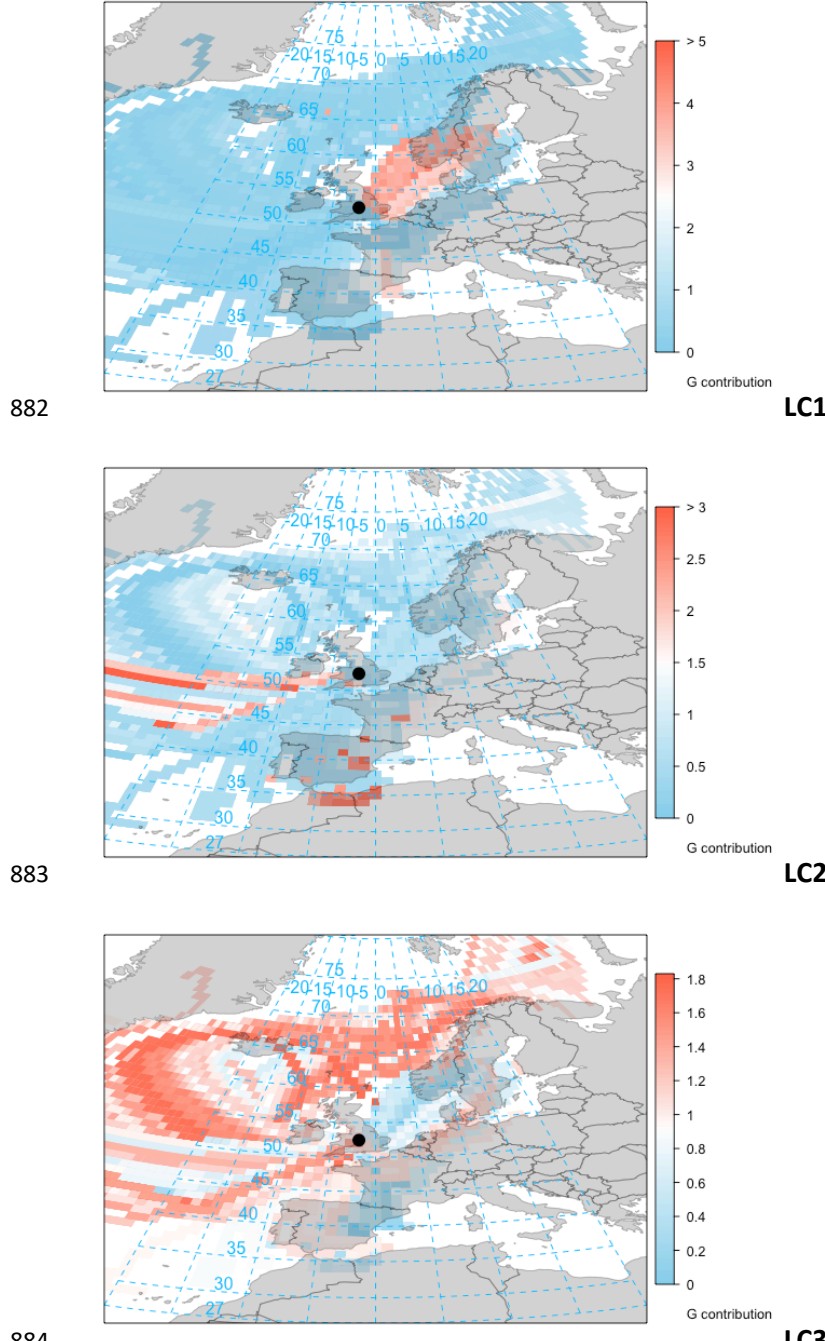

882                                       **LC1**

883                                       **LC2**

884                                       **LC3**



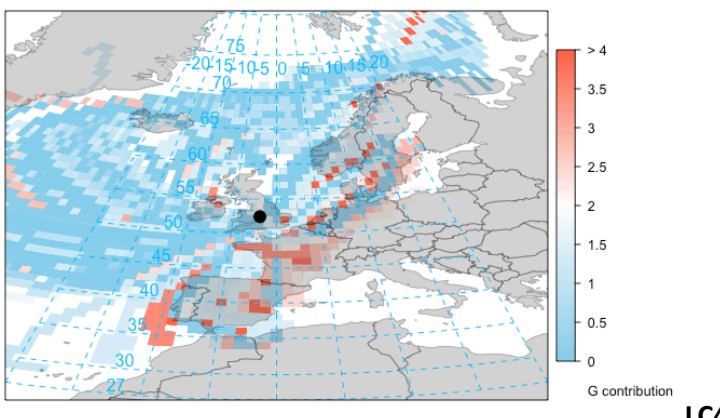

**LC4**


Figure 6: Average G contribution of the factors from the LC analysis for incoming air masses.
Higher contributions indicate better association of the given factor with the corresponding
air mass origin.







**RG1**

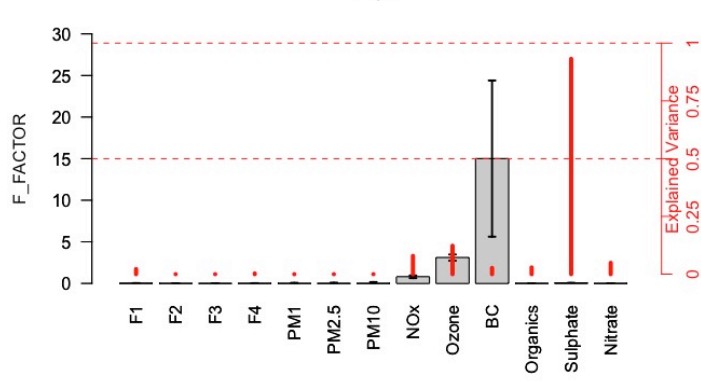


**RG2**

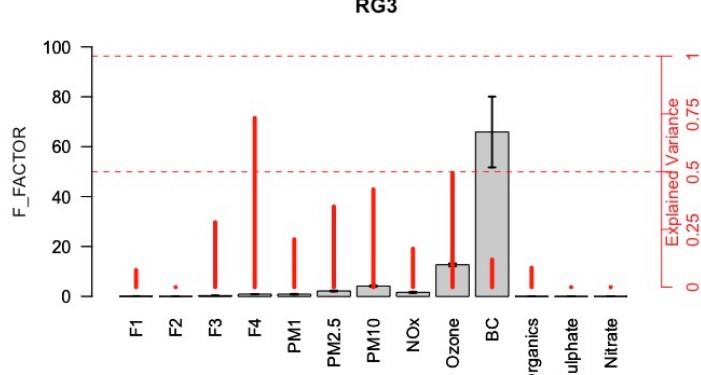


**RG3**






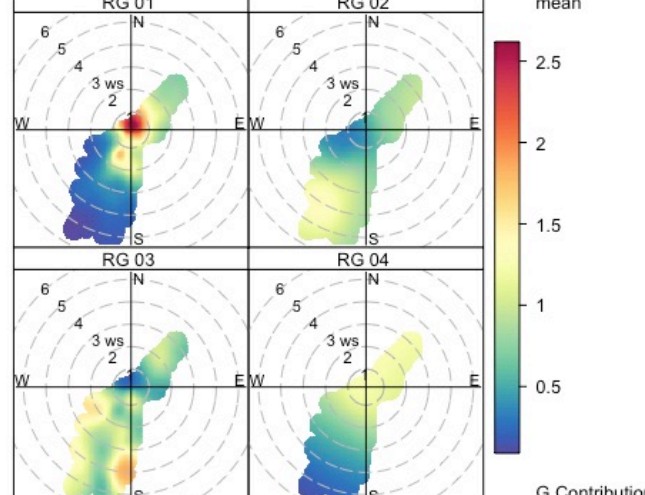


Figure 7: Variable association for the factors from the RG analysis. Grey bars indicate the
values of F, while red bars indicate the explained variations for each variable.



Figure 8: Polar plot of the average G contributions of the factors from the RG analysis.





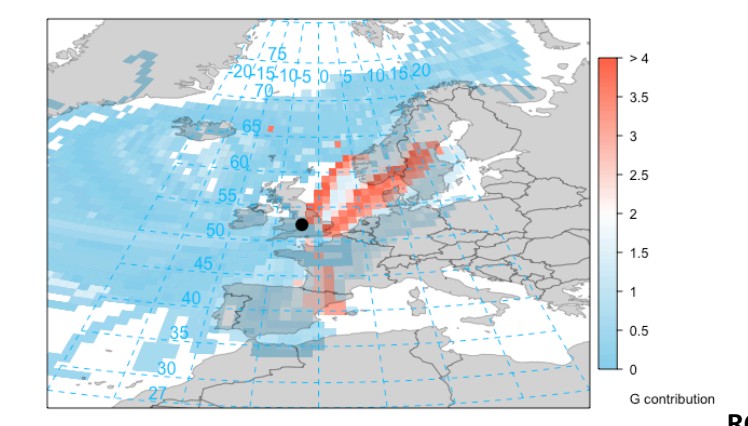


**RG1**

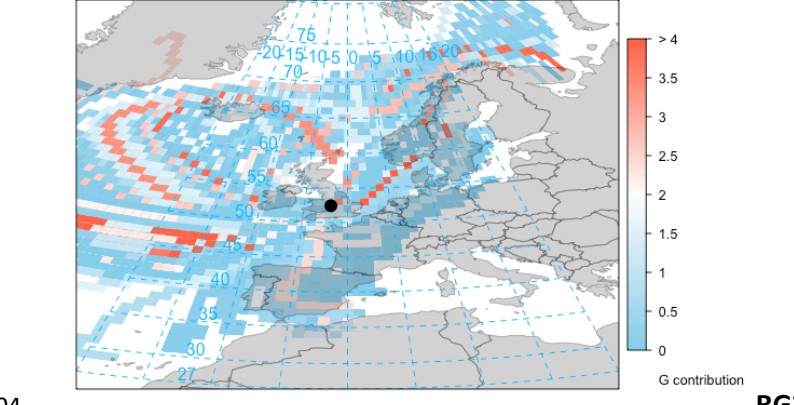


**RG2**

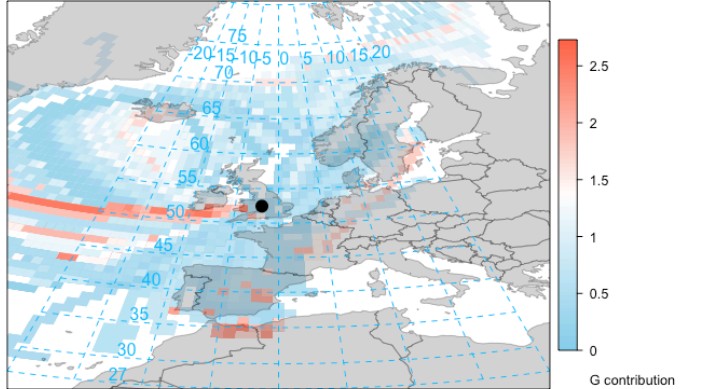


**RG3**






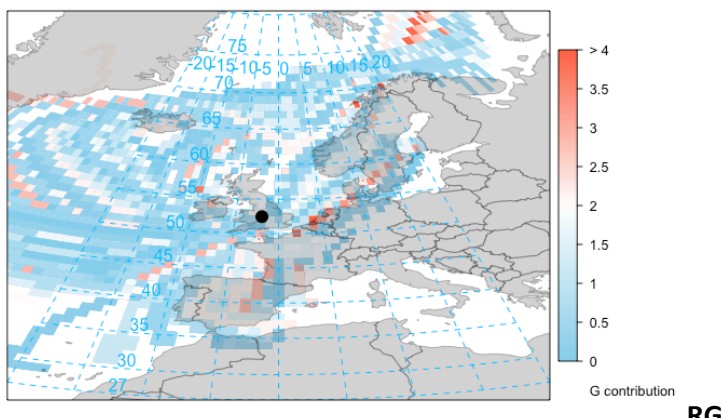

**RG4**

Figure 9: Average G contribution of the factors from the RG analysis for incoming air masses.

Higher contributions indicate better association of the given factor with the corresponding

air mass origin.