# Peer review of "A study on the performance of low-cost sensors for source"

_Atmospheric Measurement Techniques, 2022_

## Author Comment (AC2)

We thank the reviewers for their thoughtful and helpful comments. We respond to them in turn below.

**RC1**

This manuscript showcases an important method that can be applied to measurements from low-cost sensors for source apportionment. I recommend the following major revisions:

Specific comments

1) Some of the sentences in the Introduction are very long and should be shortened to make the manuscript clear

RESPONSE: The text in the abstract and Introduction was reconstructed to shorten most long sentences and remove repetitions.

2) In the methods, it might be useful to have a table detailing the different instruments, their method of operation, pollutants measured as well as if they were low-cost/ reference, and location.  This was not clear for some of the instruments mentioned, for example, the Box of Clustered Sensors. This section was a little hard to follow with the number of instruments mentioned but not described in detail.  It also wasn't clear why indicators such as LDSA were mentioned in this section and what that had to do with source apportionment. I think including a few more details about the method and the pollutants used in the Introduction would be helpful to readers.

RESPONSE: A brief description of the method of operation of each LCS was added. Additionally, a table summarising all the low cost sensors (LCS) and regulatory grade (RG) instruments used in the study, along with a clarification of their quality (either LCS or RG) and approximate cost was added in the SI. The $LDSA_{ratio}$ was used as a variable on the low-cost PMF analysis (also presented in the results in Figure 3). Its use and potential in the analysis is mentioned in the methodology as well as in the discussion (end of section 4.2)

3) The last paragraph in section 2.1 was not about the instruments at all. I suggest moving this paragraph to the next sub-section.

RESPONSE: The paragraph mentioned was moved at the end of section 2.2, which was renamed (due to this) "Positive Matrix Factorisation and data analysis"

4) When explaining the PMF method I suggest that the authors actually include equations to describe the two-step PMF process used in this analysis. The authors do not explain the limitations of using a combination of PNSD and particle composition, and the need to use the two-step PMF method. I think this is a critical point and needs to be elaborated on. How did this method differ from that used by Hagan et al.- the study the authors cited in the Introduction?

RESPONSE: Hagan used NMF, which is a version of PMF in which all components of the data matrix are weighted equally rather than with individual errors. We now highlight this on L212) "…*by Hagan et al., (2019) using Non-negative Matrix Factorisation (NMF, a version of PMF in which all components of the data matrix are weighted equally rather than with individual errors) on a dataset from New Delhi, India.*". Additionally, they included all variables in one step rather than separating them in groups and run the model in two steps (justification for this is given in the methodology). Also, their aim was not to associate PNSD profiles with pollutants (which is what is attempted in our manuscript) rather they separate sources using all multipollutant data. For the shortcomings of using different variables in a single step the following text was added in section 2.2 "Such a combination may cause several shortcomings *in the application of the PMF as different types of data are used, due to the significant difference between the nature of each variable. While this could be overcome by increasing the total weights of the primary group of measurements (the one considered better in driving the model), this could be problematic in the treatment and importance of the auxiliary dataset in the model (Beddows and Harrison, 2019)*". Finally, for better explaining how the method works the flow diagram used by Beddows and Harrison (2019), under Creative Commons privilege, along with its description is added in the SI. It is noted there are no specific equations in the process.

5) More details of the PMF method were included in the Results instead of the Methods section (eg section 3.2). This again makes it hard for the reader to follow with the authors did.

RESPONSE: The general rule for the choice of the optimal solution was moved, and edited, from section 3.2 to section 2.2. The similar point in the beginning of section 3.3 was kept though, as in this case the formation of weaker associations with the added variables is additional information specific to that analysis.

6) It appeared that without data from reference monitors, the four factors identified from the OPC data alone were hard to interpret. If so- why bother conducting a source apportionment analysis with low-cost sensors?

RESPONSE: We believe the main point of the present study is to explore the ability of LCSs alone to provide a sensible result. To do this we analysed their data alone and then compared them with those from reference monitors. This is discussed several times within the manuscript along with the limitations that come with their use. Such studies are needed to advance the use of low cost sensors, either on their own or combined with reference monitors where their weaknesses are spotted, which is crucial for the much needed denser monitoring network that can be achieved with those sensors. The argument for this need of higher density monitoring is also discussed in the text. While the results are not perfect, when compared to regulatory grade equipment, they do greatly advance the aim of increasing the ability of performing source apportionment in more situations. The limitations and prospects of the low cost approach is discussed in the Conclusions section.

7) Given that the OPCs do not measure particles < 0.3 micrometers, how useful is this technique in areas dominated by vehicle emissions?

RESPONSE: The following sentence was added at the Conclusions section to provide a response to this query to highlight the challenges that come without ultrafine particle information.

*- "though it can be challenging in sites with particle emissions smaller than what the OPC can measure, for example vehicle exhaust emissions")*

**RC2**

The authors present a new methodology for using LCS for source apportionment. This is an important topic as being able to extract source information from LCS AQ data would immensely improve the utility and power of LCS. Overall I think the paper is adequate for publication subject to minor revisions.

Specific comments:

1. I think the paper overall, but especially the abstract, could be a little more quantitative in its description. The abstract contains several instances of describing results qualitatively (e.g., "provide results that were consistent with a previous study" line 28; "good consistency between results", line 35, etc). It would be better to provide the numbers/statistics that show this rather than just telling the reader that the results were consistent.

RESPONSE: The results of source identification/apportionment studies between different methods cannot be quantitatively compared. What is compared is the consistency between the sources pinpointed and separated (according to their nature, origin, variation etc.), and this is what is discussed in the text. Thus, clarifications and notes were added to provide more explanation to the meaning of the word "consistency" in the text.

In the abstract the following sentence is updated. "*Comparing the results from a previous analysis, in which a k-means clustering algorithm was used, a good consistency between the results was found in pinpointing and separating the sources of pollution that affect the site.*"

2. There is no discussion or citation of the performance of the Alphasense OPC-N3, which is critical in interpreting the source apportionment results. Have the authors compared the PNSDs from the Alphasense to any reference field monitors or lab instruments? The performance of these optical particle counters through publicly available resources such as AQ-SPEC is fairly mixed.

RESPONSE: For the PMF model to perform well and provide meaningful results the absolute values of the variables (i.e. concentrations) are not of great importance, rather it is the relative values of the variables that is important. Regardless, the Pearson correlation coefficient is calculated between the values from LCS and RG instruments. These results are now presented in section 3.1 which is now renamed to "General conditions at the BAQS site and overall performance of the low-cost sensors" along with a note highlighting the lesser importance of the absolute accuracy of the measurements compared to the relative value among variables. Thus, the following text was added: "*Most of the LCS performed well when compared to their more expensive RG counterparts, using the Pearson correlation coefficient as the measure of correlation. The OPC-N3 presented a strong correlation for*

*PM$_1$ (r = 0.88), though its performance weakened with greater sized PM (r = 0.49 for PM$_{2.5}$ and r = 0.46 for PM$_{10}$). The decreasing correlation from PM$_1$ to PM$_{2.5}$ to PM$_{10}$ is likely due to greater wall losses in the tubing for the bigger particles. Strong correlations were also found from the BOCS sensors as well, with both O$_3$ and NO$_x$ concentrations presenting high r values when compared with their respective RG instrument measurements (0.95 and 0.82 respectively). Finally, the BC measuring LCS presented lower agreement with the measurements from the RG instrument, with a Pearson correlation value of 0.40. It is noted, in the present study the absolute performance of the LCS is not of great importance and thus it is not analysed in depth. For the PMF model to present meaningful results the representation of the relative values and variability of the variables is crucial instead, and this is thoroughly tested in the present study."*

3. Line 203 mentions separate NO/NO2 LCS data. Is this from the "Box of Clustered Sensors"? It's a little unclear what devices are being used here. I have a similar concern with the quality of the data here as well, as several studies have shown that the NO2 from alphasense gas sensors are not very reliable.

RESPONSE: In section 2.1 it is now stated that the BOCS provided both NO and NO2 measurements. Additionally, the clarification that these measurements were collected from the BOCS was also added in section 2.2. For the PMF analysis, the most important feature in providing a meaningful result is the variation of a given variable rather than their absolute values, and no quantitative results are presented in the study. This is exactly what is tested in the present study for the various LCS used. Thus, a note highlighting this point is added. The response to question 3 that the variation of a variable is the most important factor for the PMF analysis is also added in the text.

4. The data showing the source apportionment from the LCS alone (particles and gases) seems to be of weaker utility than when the ACSM is brought in. In particular LC4 does not really have any source condition associated with it, as the authors mention. I find the statement on line 461-462, saying that hyperlocal source apportionment is now possible with only LCS, to be exaggerating a little bit. I'd recommend softening that or at least adding in the caveats that some sources can't be well characterized. The way it is written now somewhat oversells the results, I think.

RESPONSE: We now include the following caveat to the text to further highlight the limitations mentioned and the potentials that will come with further work and development.

*"…though as highlighted within this study, there are some limitations for specific sources associated with pollutants with certain properties. Further exploration of these limitations and design of methodologies to overcome them, can enhance their capability and open new research and industrial abilities to pinpoint air pollution sources and subsequently manage them."*

---

## Author Response (AR2)

Dear Editor,

We have updated the references as requested.

Many thanks for your help.

Best Wishes,

Francis